# Modern Management of Esophageal Cancer: Radio-Oncology in Neoadjuvancy, Adjuvancy and Palliation

**DOI:** 10.3390/cancers14020431

**Published:** 2022-01-15

**Authors:** Francesco Cellini, Stefania Manfrida, Calogero Casà, Angela Romano, Alessandra Arcelli, Alice Zamagni, Viola De Luca, Giuseppe Ferdinando Colloca, Andrea D’Aviero, Lorenzo Fuccio, Valentina Lancellotta, Luca Tagliaferri, Luca Boldrini, Gian Carlo Mattiucci, Maria Antonietta Gambacorta, Alessio Giuseppe Morganti, Vincenzo Valentini

**Affiliations:** 1Dipartimento Universitario Diagnostica per Immagini, Radioterapia Oncologica ed Ematologia, Università Cattolica del Sacro Cuore, 00168 Roma, Italy; francesco.cellini@policlinicogemelli.it (F.C.); giancarlo.mattiucci@unicatt.it (G.C.M.); mariaantonietta.gambacorta@policlinicogemelli.it (M.A.G.); vincenzo.valentini@policlinicogemelli.it (V.V.); 2Dipartimento di Diagnostica per Immagini, Radioterapia Oncologica ed Ematologia, Fondazione Policlinico Universitario “A. Gemelli” IRCCS, 00168 Roma, Italy; stefania.manfrida@policlinicogemelli.it (S.M.); calogero.Casa@guest.policlinicogemelli.it (C.C.); viola.deluca@guest.policlinicogemelli.it (V.D.L.); giuseppeferdinando.colloca@policlinicogemelli.it (G.F.C.); valentina.lancellotta@policlinicogemelli.it (V.L.); luca.tagliaferri@policlinicogemelli.it (L.T.); luca.boldrini@policlinicogemelli.it (L.B.); 3Radiation Oncology, IRCCS Azienda Ospedaliero-Universitaria di Bologna, 40138 Bologna, Italy; alessandra.arcelli@unibo.it (A.A.); alice.zamagni4@unibo.it (A.Z.); alessio.morganti2@unibo.it (A.G.M.); 4Radiation Oncology, Mater Olbia Hospital, 07026 Olbia, Italy; andrea.daviero@guest.policlinicogemelli.it; 5Department of Medical and Surgical Sciences, IRCSS—S. Orsola-Malpighi Hospital, 40138 Bologna, Italy; lorenzo.fuccio3@unibo.it; 6Dipartimento di Medicina Specialistica Diagnostica e Sperimentale (DIMES), Alma Mater Studiorum, Bologna University, 40126 Bologna, Italy

**Keywords:** esophageal cancer, radiotherapy, guidelines, review, CTV, MR guided RT, palliation, stent, ongoing trials

## Abstract

**Simple Summary:**

Radiotherapy plays an important role in the management of esophageal cancer. Historically, it has been used in different settings—adjuvant, neoadjuvant, definitive in combination with chemotherapy, and even palliative scenario. The aim of this review is to focus on the role of radiotherapy at different levels, and to describe the new therapeutic opportunities offered by technological advances.

**Abstract:**

The modern management of esophageal cancer is crucially based on a multidisciplinary and multimodal approach. Radiotherapy is involved in neoadjuvant and adjuvant settings; moreover, it includes radical and palliative treatment intention (with a focus on the use of a stent and its potential integration with radiotherapy). In this review, the above-mentioned settings and approaches will be described. Referring to available international guidelines, the background evidence bases will be reviewed, and the ongoing, more relevant trials will be outlined. Target definitions and radiotherapy doses to administer will be mentioned. Peculiar applications such as brachytherapy (interventional radiation oncology), and data regarding innovative approaches including MRI-guided-RT and radiomic analysis will be reported. A focus on the avoidance of surgery for major clinical responses (particularly for SCC) is detailed.

## 1. Neoadjuvant Radiation Oncology

### 1.1. Current Guideline Indications

The role of multimodal treatment in esophageal carcinoma (EC), and radiotherapy (RT) in particular is well established, as showed in recent European guidelines and reviews [1]. We will refer to the most commonly referred international guidelines and table of indications [2,3].

It must be highlighted that along with the cervical, thoracic and abdominal esophagus, the primary tumor site located into the gastroesophageal junction belongs to esophagus if it can be classified as Siewert location type I and II [4,5]. Conversely, the tumor should be classified as a gastric cancer’s location if it represents a Siewert III location. The different classification of Siewert I, II or III is not always applied in clinical trials inclusion settings, often grouping together esophageal and gastric locations, thus increasing the complexity in addressing definitive conclusion on the efficacy of some therapeutic approaches for the respective clinical presentations [6]. 

The National Comprehensive Cancer Network guidelines (NCCN) [2] mainly stratifies clinical presentations by histology (adenocarcinoma (ADK) or squamous cell carcinoma (SCC)), by TNM stage and by medical fitness status for the prospect of undergoing surgery. For SCC, medically fit and locally advanced staging (within “cT2N0”-“T4a anyN”) concomitant radiochemotherapy (RTCT) either with neoadjuvant (nRTCT) or radical intent (rRTCT) is advised. If the same main presentation (i.e., SCC, medically fit) is associated with a more advanced stage (i.e., cT4b, in the setting of invasion of trachea, great vessels, vertebral body, or heart), only rRTCT or a palliative approach is suggested.

For ADK, medically fit and locally advanced staging (within “cT2N0”-“T4a anyN”), the NCCN guidelines include chemotherapy (CT) among the treatment options. It advises nRTCT (specified as the “preferred option” based on Level 1 category evidences) or rRTCT, or perioperative CT (periCT) or preoperative CT (preCT). If the same main presentation (i.e., SCC, medically fit) is associated with a more advanced stage (i.e., cT4b, in the setting of invasion of trachea, great vessels, vertebral body, or heart) only rRTCT or a palliative approach is suggested (similarly to the previously detailed presentations).

Conversely, for all the medically non-fit presentations, (SCC and ADK), NCCN advises for rRTCT, palliative RT or palliative/best supportive cares.

In 2020, the American Society of Clinical Oncology (ASCO) provided a Clinical Practice Guideline [3]. It does not differ much from the NCCN, apart from not indicating as “preferable” the nRTCT or rRTCT over periCT for operable setting of ADK presentations, and for not strictly recommending the preCT approach. Int the discussion, ASCO guidelines summarize nRTCT as preferable, particularly for bulky tumors (due to higher risk of positive resection margins) with more proximal extension, while suggesting periCT for smaller tumors located at the gastroesophageal junction without significant proximal extension, where complete surgical resection is more feasible.

In the following sections, we will look at the background evidence bases, deepen the discussion on the issue of junctional primaries (Siewert I-II), and provide an overview of the ongoing trials focused on these issues that require clarification.

### 1.2. Modern Evidence

EC is the eighth most common cancer, and is considered an aggressive disease with respect to prognosis and mortality rate [7,8].

Epidemiological changes have occurred in recent decades with an increasing incidence of ADK in distal esophagus and the gastro-esophageal junction (GEJ) in Western countries, whereas SCC remains the most common histology in Eastern Europe and Asia.

Surgery has been regarded as a mainstay of treatment for EC but the postoperative mortality [9] and higher recurrence rate with esophagectomy [10] have prompted the investigation of multimodal treatments such as periCT and nRTCT [11,12].

By pooling together all the previous trials in past decades, three meta-analyses systematically found a significant survival benefit from multimodality treatment [13,14,15].

Efficacy and safety of nRTCT has been demonstrated by a large multicenter phase III randomized CROSS trial comparing preoperative weekly paclitaxel/carboplatin concurrent with radiation therapy of 41.4 Gy versus surgery alone on 366 patients clinical stage T2–T3, N0-1 resectable esophageal or GEJ cancers [16].

This study reported a significant rate of R0 (92% versus 69% in the surgery arm; *p* < 0.001) and overall survival (OS) (49 months versus 24 months in the surgery-only arm) with low short-term toxicity and morbidity, confirming previous indications from smaller phase III studies [17,18,19] and meta-analyses [13]. This is the case since the CROSS schedule was widely adopted as one of the standards of care for patients with locally advanced resectable esophageal or junctional cancer for both SCC and AC.

The nRTCT significantly reduced locoregional recurrences from 34% to 14% (*p* < 0.001) [20] and improved overall survival at 5-year follow-up (48.6 versus 24 months in the surgery-only group (HR = 0.68; 95% CI, 0.53–0.88; *p* = 0.003) [21].

The long-term outcome analysis confirmed a persistent improvement in overall survival at 10-year follow-up, showing that nRTCT reduces the risk of dying from esophageal cancer without increasing the risk of toxicity-related death [22].

Despite encouraging results, systemic recurrence rates remain high and 5-year survival rates rarely exceed 40% [23,24], so optimizing treatment by identifying better chemotherapeutic agents and tailoring treatment to EC histological subtypes remains a priority.

Preliminary phase II trial had shown the efficacy of preoperative FOLFOX combined with RT for clinically staged II or III esophageal ADK [25].

Ongoing randomized trials will address the optimal chemotherapy schedule to combine in concomitant radiochemotherapy (Folfox to paclitaxel and carboplatin PROTECT trial) both with RTCT (41.4 Gy), in patients with resectable stage IIB–III esophageal and GEJ cancers of SCC or ADK histology [26].

Extending the inclusion criteria for nRTCT to clinically resectable and locally advanced EC (cT1/N+ or cT2-4a/N0-3/M0) with weight loss > 10% and/or age > 75 years, the surgical radicality rate and pCR remain high but with increased postoperative mortality and morbidity [27].

This might indicate that other therapeutic options such as rRTCT could be considered, as it has been shown to be an alternative for patients who are not fit enough to be treated with nRTCT [28].

In this context, RTCT has historically been advantageous in terms of both survival and local failure compared to RT alone [29].

From the perspective of personalization of treatments, the possibility of surgery in selected patients achieving complete response after rRTCT was evaluated. Although the study did not reach the goal of improving 1-year OS, it was found that an rRTCT approach is preferable to surgery in patients with SCC of the cervical esophagus due to the increased morbidity rate associated with the surgical procedure [30].

Regarding the combination of drugs used, the NCCN panel recommends regimens with Paclitaxel and Carboplatin as the preferred approach for rRTCT. Although not advantageous in terms of PFS, results from the PRODIGE5/ACCORD17 trial described the feasibility of rRTCT with the FOLFOX scheme in patients not eligible for surgery [31].

The effectiveness of nRTCT in patients with esophageal SCC is controversial and several studies investigating the curative potential of RTCT have challenged the idea that surgery is an indispensable part of curative therapy [32,33].

Compared to esophageal ADK, patients with SCC tend to gain more benefits from nRTCT, which was confirmed by a NEOCRTEC 5010 study [34].

A meta-analysis of randomized controlled trials compared chemoradiation plus surgery with RTCT alone in patients with at least T3 and/or N+ thoracic esophageal cancer (93% had SCC) concluded that the addition of surgery to chemoradiation in locally advanced esophageal SCC has little impact on OS, and may be associated with higher treatment-related mortality [35].

The results of this meta-analysis do not change clinical practices but allow us to generate future research hypotheses.

A further aspect to be investigated is the use of periCT in SCC; in fact, although the CROSS study tends to be more effective in SCC, periCT is indicated by Japanese guidelines as the preferred choice after the results of the JCOG9907 study [36,37].

An ongoing three-arm NExT tudy (JCOG 1109) is comparing nCF with nCF + Docetaxel and nCF with 41.4 Gy RT for SCC, with the ultimate goal of providing evidence for the superiority in efficacy of one of these options [38].

The investigation of the optimal CT schedule and the radiation dose in nRTCT remain to be further investigated, and there is room for improvement in outcomes with the contribution of modern radiotherapy not only in the complexity of the treatment planning and delivery, but also in the understanding of the biologic processes that underline the radiation responses [6,39,40].

For approaches to treatment in the future, it is also crucial to consider the molecular profile of the disease and include the more promising new molecular therapies. Studies dealing with molecular targeting agents (e.g., EGFR, VEGF and HER) in concomitant combination with RT are ongoing and specifically recruiting patients with GEJ lesions [41].

### 1.3. The Issue of Junctional Primaries: In Other Words, Shall We Still Prefer RTCT over CT for GEJ?

The most common site of esophageal cancer is the distal site, which often involves the GEJ.

Anatomically, the GEJ separates the lower esophagus from the proximal stomach, typically in the area where the squamous epithelium of the esophagus changes into the columnar epithelium of the gastric cardia [42]. Adenocarcinomas of the GEJ represent around 90% of all GEJ cancers [43]; they are generally considered less radiosensitive than squamous cell carcinomas.

Siewert classification defined three types of GEJ lesions according to the localization of the lesion’s epicenter on the range of distance from the GEJ (more than 1 cm above; between 1 cm above and 2 cm below; over 2 cm below that) [44,45].

Type I and II belong to esophageal cancers, while type III belongs to gastric primary, as it is by current TNM classification [46]. Surgical series described different survival trends between the three subtypes, with better outcome for types I and II compared to type III. Moreover, they described the nodal spread patterns associated with each type, which are particularly useful to guide both surgical nodal dissection and RT planning (in terms of prophylactic nodal target definition of areas to be irradiated as clinical targets) [15,46,47]. The optimal treatment strategy for locally advanced ADK of the GEJ is still under discussion.

The most definitive results in support of the use of nC nRTCT for patients with GEJ cancer come from the CROSS trial, in which a high proportion of esophageal ADK (75%) was analyzed; nRTCT could confer a better local tumor control with improved R0 resection rates, higher pCR rate, and fewer lymph node metastases compared with periCT, but no survival difference was observed between preoperative RTCT and CT [48,49,50].

Recently, the meta-analysis published in 2018 by Zhao analysing six studies (total of 866 pts—50% ADK) suggested that nRTCT should be preferred to periCT with a significant long-term survival benefit in patients with EC or the GEJ and a statistically significant difference between in the incidence of postoperative complications such as pulmonary, anastomotic leak and cardiovascular complications without significant differences in perioperative mortality (RR = 1.85, 95% CI = 0.93–3.65, *p* = 0.08) [51].

A propensity score matching analysis on 170 pts with Siewert II and III adenocarcinoma nRTCT confers a better survival with improved R0 resection rate and pCR rate compared with neoadjuvant CT with no significant increase in postoperative complications for the patients with locally advanced adenocarcinoma of GEJ [52]. On the other hand, some evidence advocated periCT as preferred modality for ADK histology.

The FLOT trial showed that patients with gastric or GEJ ADK had better overall survival when treated with perioperative fluorouracil, leucovorin, oxaliplatin, and docetaxel (FLOT), with 2-year and 5-year overall survival of 65% and 39%, respectively, similar to the CROSS trial but burdened by higher rates of severe toxicity [53]. Ongoing trials will help to identify the best therapeutic approach (nRTCT versus periCT), their integration (pCT +/− followed by RTCT), and optimal CT schedule for different EC presentation (Siewert type and histological subtype).

Ongoing neoadjuvant trials for ADK: ESOPEC (cross versus Flot)-NEOAEGIS (flot versus CROSS) directly compare CROSS and FLOT for AD; RACE (FLOT versus FLOT + RCT).

Until then, nRTCT will be the preferred strategy for SCC and ADK Siewert types I and II for at least two reasons; the first is that 75% of the patients in the CROSS trial had an ADK histology, and the second is because the benefits of periCT have been derived from studies conducted on patient populations with both stomach and esophageal cancer, making it questionable to apply to esophageal ADK presentation.

### 1.4. Doses and Volumes (Including Brachytherapy/Interventional Radiation Oncology)

In the main randomized clinical trials regarding neoadjuvant chemoradiation for EC, the prescribed doses were 41.4 Gy in 23 daily fractions of 1.8 Gy in the CROSS trial [16]; 50.4 Gy (45 Gy on the largest volume and 5.4 as a boost) in 28 daily fractions of 1.8 Gy in the CALGB 9781 trial [19]; 40 Gy in 20 daily fractions of 2 Gy in the NEOCRTEC5010 trial [34]. In those trials, clinical target volume (CTV) was defined as the extension of the primary gross tumor volume (GTV) with a proximal, a distal and a radial margin of 3–5 cm, 3–5 cm and 0.5–1.5 cm, respectively, with the inclusion of elective nodes area.

Modern radiotherapy, with Intensity Modulated Radiation Therapy (IMRT), and the eventual use of Simultaneous Integrated Boost (SIB), provide an opportunity to safely perform dose escalation, as reported by Innocente et al. [54], who proposed a dose prescription up to 52.5–55 Gy in 25 daily fractions. In another study, Lo et al. [55] reported that higher doses (50 Gy versus 36 Gy) have increased pathologic complete response (pCR) rate and overall survival outcome.

The dose–response relationship between EC and pCR is not unequivocally demonstrated: on the one hand, results of a meta-analysis indicate that higher doses led to higher pCR rates [56]; however, in other studies, Yang et al. [57] reported that higher doses (>45 Gy versus ≤45 Gy) were not associated with higher pCR rate or with a longer survival and, in a recent systematic review with a pooled analysis [58], the best biologically equivalent dose (BED) identified by the authors was BED ≤ 48.85 Gy, versus BED > 48.85 Gy.

Information regarding dose–response attitudes of EC could be found in studies that explored the role of radiotherapy in definitive settings: in fact, in a dose escalation trial [59], Zhang et al. reported that the SIB treatment prescribed with 63 Gy on the GTV and 50.4 Gy to subclinical disease in 28 daily fraction was well tolerated and offered interesting long-term survival.

Moreover, in the definitive setting, interventional radiotherapy (IRT, brachytherapy) is also currently proposed as a promising technique to boost primary GTV and allow a safe dose escalation [60] or, in selected patients affected by early-stage EC, as exclusive treatment [61]. High dose rate (HDR) IRT is usually proposed with a total dose of 10–36 Gy (10–15 Gy as a boost after chemoradiation or 25–36 Gy as exclusive treatment [61], with a dose per fraction that should not exceed 5 Gy [60]). Prospective studies with larger case series are required to confirm the efficacy and the safety of dose-escalation, combining external beam radiotherapy and interventional radiotherapy.

### 1.5. Ongoing Trials

Trials on neoadjuvant RTCT versus perioperative CT.

The German ESOPEC trial is a multicenter, prospective, randomized controlled two-arm trial comparing nRTCT to periCT followed by surgery, in terms of OS in localized esophageal adenocarcinoma [62]. In the experimental arm, patients undergo four preoperative and four postoperative CT cycles according to FLOT protocol [53].

In the comparative arm, patients are randomized to receive RTCT according to CROSS protocol followed by surgery [21]. In the multicentric phase III randomized trial, Neo-AEGIS, two established treatment protocols are compared in terms of survival benefit in esophageal and GEJ adenocarcinoma [63].

Eligible patients are randomized in a 1:1 design between the modified MAGIC regimen [64] or multimodality RTCT according to CROSS protocol [21]. The randomized phase III RACE trial is investigating progression-free survival in multimodality treatments for potentially resectable adenocarcinoma of the GEJ [65]. In the experimental arm, preoperative induction FLOT CT is followed by nRTCT (45 Gy RT with weekly oxaliplatin plus 5-FU); while in the control arm, patients undergo four cycles of preoperative FLOT CT alone. In both arms, patients undergo resection and four cycles of postoperative FLOT.

#### Trials on Preoperative RTCT Dealing with Choice of Which Schedule of Concomitant CT

The French Protect Trial, a prospective multicentric randomized phase II trial, is evaluating the short-term complete resection rate and safety of two different concomitant CT schedules in nRTCT for operable esophageal and junctional (Siewert I-II) cancer [26].

Ongoing trials in neoadjuvant setting are summarized in Table 1.

## 2. Adjuvant Radiation Oncology

### 2.1. Current Guideline’s Indications

Recently published ASCO guidelines do not deepen the issue on how deal with postoperative management for esophageal presentations [3]. NCCN guidelines approach the scenario by histology (SCC or ADK) and whether patients underwent a preoperative approach or not [2]. Approaches are shuffled on the basis of microscopically complete (R0), incomplete (R1) or macroscopically incomplete (R2) resection.

In case of patients not having received a preoperative approach, the approach is overall more aggressive; for both SCC and ADK, in case of R1-2, RTCT is indicated (or a palliative approach for more advanced R2). For R0 cases, while SCC are recommended to only be surveilled, for ADK R0, in case of pT2-T4a, a RTCT can be considered; however, for R0 ADK pTis or pT1, only surveillance is needed.

In case of patients having received a preoperative approach: for both SCC and ADK, in case of R1-2, observation or palliative care are indicated (with the exception of the chance for re-resection if possible, for ADK R1 only). In case of an SCC R0: surveillance or nivolumab is indicated. In case of an ADK R0, the approach is broader: for pathological complete responses, only observation or the completion of the periCT (if preoperatively started) is indicated; conversely, for pTpositive/Npositive R0 ADK, aside from to the previous two options, the administration of Nivolumab is also suggested. ASCO recently developed guidelines regarding the integration of immunotherapy into the treatment management of esophageal cancer [66].

### 2.2. Available Evidence

Postoperative irradiation has historically been reserved for patients who had bulky tumors with gross residual/histologically proven microscopic residual. In fact, the main advantage of the adjuvant approach versus the neoadjuvant approach is the knowledge of the pathologic staging; in fact, adjuvant therapy may allow us to treat areas at risk for recurrence. Conversely, patients with pathologic T1, N0, and M0 or metastatic disease may be spared treatment.

The potential disadvantages of adjuvant RT include limited tolerance of normal tissues that have already been challenged by post-surgical changes (such as fibrosis or adhesions), potentially wider fields compared to preoperative, and a potential delay in adjuvant treatment administration.

Despite the potential disadvantages outlined above there is robust data in the literature to support the importance of radiation therapy in the treatment of esophageal cancer in the adjuvant setting [67,68,69].

Several trials have investigated the role of surgery followed by postoperative RT versus surgery alone, both in ADK and SCC, with no significant survival differences and better locoregional recurrence in the RT arm (70% versus 85%). Moreover, locoregional recurrence was significantly improved in patients without nodal involvement who received adjuvant treatment (65% versus 90%) [70]. For patients with involved lymph nodes, 5-year survival rates for surgery alone patients versus patients receiving resection and RT were 17.6% and 34.1%, respectively (*p* = 0.04) [71].

On the other hand, adjuvant RT was also associated with increased morbidity and death, as well as the early appearance of metastases (an aspect that might be related to the high dose per fraction and large total dose delivered) [72].

In conclusion, adjuvant RT may decrease local recurrence, especially in cases of close/positive margins, T3, N+, lymphovascular/perineural invasion, and poorly differentiated tumors, although its real impact on overall survival still remains less clear [73].

Adopting a combination treatment of radiotherapy and chemotherapy has also shown its advantages in the adjuvant setting, even if current guidelines generally recommend RTCT in the neoadjuvant setting [74].

Data from the Taiwan Cancer Registry database showed a better 3-year survival rate for patients who received adjuvant RTCT in comparison to those who only underwent surgery (44.9% versus 28.1%); once again, the following factors were statistically significant predictors for clinical outcome: ypT3-T4, tumor length > 32 mm, ypN+, and evidence of either microscopical or macroscopical residual at resection (R1 or R2). It is, therefore, evident that RTCT offers the benefit of compensating for those adverse features individuated after surgery [75].

### 2.3. Ongoing Trials and Promising Strategies

Future perspectives in the adjuvant management of esophageal cancer may certainly include proton therapy, as it allows for a rapid drop in the dose delivered in the immediate vicinity of the target volume, allowing potentially better sparing of neighboring healthy organs, even while irradiating large anatomical districts. The dosimetric benefit of the application of proton therapy has been investigated in several retrospective case studies comparing it with IMRT. In particular, with the implementation of pencil beam scattering PBT (PBS-PBT), i.e., proton-modulated intensity, a dosimetric benefit in dose sparing to the heart and lungs has been demonstrated [76].

Currently, there is very limited clinical data available. Initial case reports have investigated the feasibility of the technique and the potential association with post-surgical complications. Compared with other techniques, an improved trend in the reduction in postoperative pulmonary, cardiac and wound complications has been reported [77,78].

Furthermore, the reduction in total body dose provided by PBT has been found to be associated with a reduction in grade 4 lymphopenia, which has a negative impact on all survival outcomes in esophageal cancer patients [79]. This evidence reported in the literature provides the basis for prospective studies to validate the feasibility of PBT and intensification of treatments with dose escalation protocols and combination with systemic treatments [80].

The MD Anderson Cancer Center group is enrolling patients in order to investigate which strategy is more effective and safer between proton therapy and IMRT, both in terms of progression-free survival (PFS) and total toxicity burden, i.e., composite score from serious adverse events and, among patients who undergo surgery, postoperative complications (ClinicalTrials.gov n°: ClinicalTrials.gov Identifier: NCT01512589).

Recent trials have shown the feasibility and positive impact on outcomes of integrating new drugs (i.e., immunotherapy) in the treatment of patients affected by esophageal and GEJ cancer. Trastuzumab is suitable for use in patients with HER2-positive EGJ adenocarcinoma in first-line treatment in combination with fluoropyrimidine and platinum-based chemotherapy [81].

Pembrolizumab is a PD-1 monoclonal antibody that has been approved by the Food and Drug Administration (FDA) for second-line use in patients with advanced or metastatic esophageal cancer [82]. The recent KEYNOTE-590 study showed promising results regarding the use of Pembrolizumab as first-line treatment combined with first-line chemotherapy in GEJ and esophageal cancer [83]. Ramucirumab, a VEGFR-2 antibody, has been approved for the treatment of EGJ adenocarcinoma and stomach refractory, or for disease progressing beyond first-line chemotherapy [84]. Recently, the FDA also approved the use of the PDL-1 agent Nivolumab for the treatment of advanced esophageal SCC as an effective treatment option for patients previously receiving chemotherapy [85].

In addition, an integrated treatment strategy between radiotherapy and new drugs (i.e., immunotherapy) is becoming more and more popular, with the dual purpose of enhancing the cytotoxic effect of irradiation and, at the same time, stimulating an immune response of the body directed towards the tumor, with a significant benefit at the systemic level. The new drugs under study for esophageal cancer in combination with radiotherapy are Cetuximab (anti-EGFR) and anti-PD1 antibodies such as Camrelizumab (ClinicalTrials.gov n°: NCT04741490). Another aspect worth investigating concerns radiation dose-escalation. Indeed, after reviewing the results for locoregional relapse according to the dose and the recommended volumes, there is a growing need to understand why increasing the dose of radiation has no impact in esophageal cancers, by trying to increase the dose administered not only on the surgical bed but also on the lymph node stations most at risk of tumor spread (French randomized phase II/III trial by ClinicalTrials.gov n°: NCT01348217).

Ongoing trials are briefly summarized in Table 2.

## 3. Palliative Approaches

### 3.1. Current Guideline’s Indications

For both SCC and ADK, in case of unresectable locally advanced, locally recurrent, or metastatic presentation, NCCN guidelines similarly indicate CT or palliative medical care on the basis of the patient’s performance status. The goal of best supportive care is to prevent and relieve suffering arising from the presence of the disease and to support the best possible quality of life. For esophageal cancer, interventions undertaken to relieve major symptoms may result in a significant prolongation of life; a multimodality interdisciplinary approach to palliative care of the esophageal cancer patient is encouraged [2]. The addition of systemic chemotherapy to best support care may not only increase survival but also improve quality of life and should, therefore, be considered according to the patient’s ECOG PS (Eastern Cooperative Oncology Group performance status). For selected cases at high logistic complexity, including patients with COVID-19 or living far away from the Radiotherapy center, or patients who are about to be admitted in long-term hospice but have a chance to receive an advantage by symptom relief: dedicated guidelines providing alternative treatment schedules and forms to perform telemedicine remote evaluation of the patient [86]. As an example, alternative radiation oncology approach has been proposed to shorten the time needed for treatment delivery: accelerated hypofractionated schedules can effectively provide symptom relief, avoiding a significant increase in the related toxicity. This has been described as effective and applicable with less advanced technology; this issue is of particular relevance for the related opportunity to offer symptom relief by esophageal cancer in developing countries [87].

### 3.2. Use of Stent and Potential Integration with Radiotherapy

Esophageal stent placement represents a palliative approach in patients with dysphagia due to inoperable esophageal cancer; in particular, stenting should be considered in those patients with an expected short-term survival, because of its rapid relief of symptoms [88]. This approach should be preferred over laser therapy, photodynamic therapy and esophageal bypass. The combination of external beam radiation therapy (EBRT) and self-expanding metal stent (SEMS) placement has been associated with prolonged dysphagia relief and improved overall survival [89]; however, a high risk of major adverse events, such as perforation and fistulas development, has been also reported. Therefore, stent placement should be better considered for patients who have failed prior radiotherapy.

Nevertheless, SEMS placement with concurrent single-dose brachytherapy has been demonstrated to be safe and effective [90]. Therefore, the use of irradiated SEMSs, which can potentially combine the advantage of SEMS placement and brachytherapy, has also been investigated. A meta-analysis of six RCTs suggested that irradiated SEMS obtained a prolonged dysphagia-free time when compared with traditional SEMS, without significantly increasing the adverse event rates [91]. Of note, all the randomized clinical trials have been performed in Chinese populations; therefore, prospective studies in Western populations are needed before any definitive suggestion can be made on the use of irradiated stenting in the palliative approach of patients with dysphagia due to esophageal cancer.

### 3.3. Clinical Application of Brachytherapy (Interventional Radiotherapy)

Interventional radiotherapy (IRT, better known as brachytherapy) is one of several RT techniques available with the potential for improving the therapeutic ratio due to the delivery of a high dose within the target volume, rapid dose fall-off in adjacent organs at risk, short treatment time, and good functional outcomes [92,93]. More frequently esophagus IRT has been utilized as a boost after EBRT showing a median local control (LC), disease free-survival (DFS), OS and cancer-specific survival (CSS) of 77% (range 63–100%), 68.4% (range 49–86.3%), 60% (range 31–84%), and 80% (range 55–100%), respectively, and a grade 3–4 toxicity range of 0–26%.

Comparing these results with surgery, the 5-year OS was lower than surgery (65–100% versus 60–84%). The discrepancy in 5-year OS between surgery and IRT could be due to patient’s features. Patients who underwent IRT usually presented a higher rate of comorbidities compared to those who underwent surgery. Probably, in patients without several comorbidities, the 5-year OS rate might be improved. To endorse this consideration, the median 5-year CSS rate is 80% (range 55–100%). These values are better than median 5-year OS and are comparable to those obtained with surgery [61,94,95].

Regarding palliation, IRT has been shown to provide more effective and longer relief (68% of patients experienced complete resolution of dysphagia) than other procedures (IRT group median dysphagia-free survival of 99 days versus 35 days) [96]. Furthermore, serious side effects rarely occurred [97].

Two randomized controlled trials compared IRT with endoluminal stent showing more effective and more durable relief from dysphagia in the IRT group and in patients who survived more than 6 weeks [98,99,100,101].

The role of EBRT in association to IRT is controversial. The only two randomized studies that compared IRT alone to IRT plus EBRT for the management of malignant [102] dysphagia showed contrasting results [103].

## 4. Innovative Approaches

### 4.1. Is There Room for Avoidance of Surgery in Case of Major Response?

In patients with resectable EC, nRTCT provides improved OS and DFS compared to neoadjuvant CT [104] or surgery alone (SA) [105,106]. In particular, the long-term analysis of phase II/III trials on advanced EC carried out at MD Anderson [104] reported not only significantly improved OS (*p* = 0.046) and DFS (*p* = 0.015), but also a significantly higher pCR rate after nRTCT compared with CT (29% versus 3%, respectively; *p* < 0.001). Moreover, RTCT significantly improved the nodal downstaging (*p* = 0.001). Nevertheless, a subsequent subanalysis of a meta-analysis [14] comparing 99 and 98 patients treated with preoperative RTCT or CT, respectively, did not find any significant advantage in terms of all-cause mortality.

Results from the CROSS trial [16] demonstrated a higher R0 resection rate in ECs treated with RTCT plus surgery versus SA (92% versus 69%, *p* < 0.001), with 29% pCR rate after the trimodal approach. Particularly, the pCR rate was 23% versus 49% (*p* = 0.008) in ADK and SCC, respectively. Moreover, a long-term update of the CROSS trial [21] showed significant benefits in terms of OS for SCC-receiving the 3 treatment modalities versus SA (median OS: 81.6 versus 21.1 months, respectively; *p* = 0.008).

The independent prognostic impact on OS of major responses after RTCT has been shown in some studies on EC [107,108]. In addition, a major response after RTCT allows the selection of patients with better prognosis and with fewer likely benefits from post-RTCT surgery. Furthermore, a 72.7% pCR rate in patients with complete clinical response (cCR) after RTCT was reported in one study [109]. Moreover, clinical response was correlated with significantly improved 3-year OS compared to non-responder patients.

Nevertheless, some differences in terms of pattern of failure and response after RTCT were reported based on EC histology. In particular, with SCC being more sensitive than ADK to RTCT, its management should be different. In fact, a trimodal strategy remains the ADK standard of care, but is debated in SCC.

More specifically, avoiding surgery seems preferable in cervical ECs based on (i) the efficacy of definitive RTCT in this anatomical site, (ii) the higher cCR rate recorded in SCC (histological type prevalent in cervical ECs), and (iii) the efficacy of PET scanning in the active surveillance of patients with cCR. Moreover, RTCT avoids the perioperative and long-term morbidity of the trimodal treatment as reported by two meta-analyses [110,111].

In patients treated with RTCT alone, salvage surgery should be reserved to locoregional recurrences based on studies showing no differences in terms of OS or PFS compared to the trimodal strategy [112].

However, future trials should further test the watchful waiting approach after RTCT to allow for a more personalized treatment strategy through reserving salvage surgery only for patients with locoregional relapse.

A phase III multicenter, stepped-wedge, cluster randomized, controlled, non-inferiority trial is investigating if active surveillance can lead to non-inferior OS and improved quality of life and cost-effectiveness compared to standard esophagectomy [113].

### 4.2. May Definitive RTCT Replace Surgery for SCC?

Combined modality treatments remain the preferred treatment option of locally advanced SCC (LAESCC). In fact, international guidelines [2,3] recommend definitive RTCT in cervical LAESCC and in patients unfit for surgery or declining surgery.

The synergistic effect of concurrent RTCT improves early treatment of micrometastases and tumor downstaging, with potentially higher resectability rates. Moreover, combined RTCT might also improve the therapeutic outcomes, particularly in the highly radiation- and chemo-sensitive SCC setting [111].

Since early 2000, several randomized controlled trials compared RTCT followed by surgery and RTCT alone [32,33]. The long-term (median follow-up: 10 years) results of a phase II trial investigating RTCT (total radiotherapy dose: 40 Gy) followed by surgery versus rRTCT alone (total radiotherapy dose: 65 Gy) in LAESCC reported no significant differences in terms of 5- and 10-year OS between the two arms despite improved local control in the surgery arm. Moreover, at multivariable analysis, cCR after RTCT was independently correlated with improved OS (HR: 0.30, 95% CI 0.19–0.48) [114].

These findings are consistent with the results of a randomized trial by the Fédération Francophone de Cancérologie Digestive [33]. In fact, Bedenne et al. compared nRTCT followed by surgery to rRTCT in LAESCC and reported similar OS and quality of life among the two arms. More recently, a Cochrane review [35] including 431 participants (LAESCC: 93%) suggested that post-RTCT surgery, compared to RTCT alone, is correlated with higher treatment-related mortality rates (RR 5.11, 95% CI 1.74–15.02; *p* = 0.003), without significant OS improvement (HR 0.99, 95% CI 0.79–1.24; *p* = 0.92).

Finally, a non-surgical conservative approach based on rRTCT alone should also be considered in potentially resectable LAESCC, particularly in patients with cCR after RTCT.

### 4.3. Role of MRIgRT

Magnetic resonance hybrid linear accelerators (MR-Linacs), combining 1.5 T or 0.35 T MRI, have recently been introduced in radiotherapy to perform magnetic resonance-guided radiotherapy (MRgRT) [115]. In esophageal cancer, MRgRT can improve visualization of the target and OARs. Furthermore, gating protocols could help mitigate respiratory motion through direct tumor tracking using real-time online imaging, especially in esophageal junction tumors [116,117]. Online adaptive (OA) radiotherapy protocols could allow for treatment plan adaptation based on the patient’s daily anatomy, aiming to customize treatment strategy by monitoring response to therapy [118,119,120]. The daily monitoring of response to therapy may also allow for tumor reduction, which occurs in 28% of patients with cancer of the esophagus from the fifth week of treatment, to be followed and managed through OA [121]. These opportunities increase the precision of treatment, reducing CTV-PTV margins and enabling safe dose-escalation strategies, in order to increase disease control. Smaller margins may allow for a reduction in the dose to OARs and, therefore, toxicity, such as to the heart, stomach and lungs, which may adversely affect treatment compliance [116]. Future trials are needed to validate the benefits of MRgRT in esophageal cancer, such as the implementation of the dosimetric benefits of MR-integrated proton therapy (MRIiPT) in a clinical setting [122].

### 4.4. Role of Radiomic Analysis

Accurate staging, treatment planning and prognostication are very important points to be considered in the treatment of EC patients. Therefore, researchers turned their attention to innovative emerging applications such as radiomics with the use of non-invasive imaging techniques in order to improve patient outcomes. Different image modalities such as computed tomography (Ct), positron emission tomography-Ct (PET-Ct), magnetic resonance imaging (MRI), and endoscopic ultrasonography (EUS) are commonly used for staging and follow-up in order to access previously hidden information, which may provide insight into the pathogenesis and comprehensive characterization of EC [123]. Ct and PET are the most frequently used techniques in the treatment of EC. However, their reduced accuracy in describing small-sized lesions is a major setback, thereby limiting the sensitivity and specificity of these imaging modalities [124].

Interestingly, the development of Machine Learning (ML), a subset of Artificial Intelligence (AI) with the ability to learn and predict outcomes from large data sets without explicit programming, has provided new possibilities in image analysis [125]. Nonetheless, questions about its interpretability and clinical value became the main focus of many studies. In an attempt to answer these questions, Xi et al., conducted a comprehensive retrospective review of studies based on ML techniques using non-invasive medical imaging in EC patients, to evaluate relevant clinical objectives such as treatment response, prognostication prediction, diagnosis, and biological characterization [126]

Different advanced image analysis techniques have been utilized so far for EC. Results showed that ML models achieved reliable predictions in the evaluation of treatment response and significantly improved the evaluation of the assessment of regional lymph nodes (N status) [127,128,129]. However, some possible drawbacks were also noted. In some of the studies, the ML algorithms used were trained and validated on the same dataset, thereby leading to inaccurate estimations of ML models that should undergo more complete validation steps. Other limitations included unreliability in the proposed models due to inadequate sample size in some of the studies and different reference standards for each outcome, thereby affecting comparison among the groups [126].

Promising results obtained from these recent studies demonstrate the effectiveness of AI-based imaging analysis to set up reliable clinical decision support tools in the treatment of EC patients and foster future prospects in image analysis. However, improved study designs will help clarify uncertainties about the real translational value and clinical significance of these observations.

### 4.5. Prehabilitation and Rehabilitation for Esophageal Cancer

In the management of a cancer patient, a pre-habilitation path associated with a rehabilitation path is becoming increasingly important.

Cancer rehabilitation is medical care that diagnoses and treats patients’ physical, psychological and cognitive impairments, in order to maintain or restore function, reduce symptom burden, maximize independence and improve quality of life in this medically complex population.

Cancer pre-habilitation is a process on the cancer continuum of care that occurs between the time of cancer diagnosis and the beginning of acute treatment, and includes physical and psychological assessments that establish a baseline functional level, identify impairments, and provide interventions that promote physical and psychological health to reduce the incidence and severity of future impairments. Both techniques were initially successfully evaluated in the patient undergoing elective orthopedic surgery. The first studies showed that in the patient undergoing elective orthopedic surgery, performing post-surgery rehabilitation allowed for a more rapid recovery of functions [130]. The addition of a pre-habilitative path also allowed improvement in physical performance, such as reducing complications related to surgery and presenting a reduction in post-surgical performance with a better result than the pre-surgical one of those subjected only to post-treatment rehabilitation.

The concept of pre-habilitation and rehabilitation linked to active oncological treatments has been developing in recent years. It has highlighted how a correct nutritional approach associated with a pre-habilitative and rehabilitative path substantially improves performance and response to oncological treatments.

This scenario is more valid in the case of an elderly cancer patient, especially if subjected to treatments on esophageal cancer, in which physiological changes related to age must be associated with the consequences of oncological therapies. In an elderly patient, we see a progressive physiological reduction in lean muscle mass and increased fat mass with an associated decrease in muscle quality [131]. This picture frequently evolves into sarcopenia with an associated increase in toxicity and susceptibility to adverse outcomes. The improvement in pre-treatment performance is associated with a reduction in perioperative complications and a lower risk of developing a sarcopenic picture in the continuation. Concerning the timing, the process does not delay treatments but uses the dead times before surgery and waiting for any systemic treatments

## 5. Conclusions

The modern management of esophageal cancer is crucially based on a multidisciplinary and multimodal approach. Radiotherapy is involved in neoadjuvant and adjuvant settings; moreover, it includes radical and palliative treatment intention. Modern radiotherapy applications and the growing role of immunotherapy will lead to optimized clinical results even in complex clinical scenarios.

## Figures and Tables

**Table 1 cancers-14-00431-t001:** Design and main characteristics of ongoing trials for pre-operative radiochemotherapy in esophageal and gastroesophageal cancer. (Abbreviations: CT: Chemotherapy; RT: Radiotherapy; CRT: Chemoradiation; OS: Overall Survival; PFS: Progression-Free Survival; CRR: Complete Resection Rate).

Trial	Clinical Subset	Study Design	Arm A	Arm B	Estimated Enrollment	Primary Endpoint
**ESOPEC** [62]	Esophageal Adenocarcinoma Adenocarcinoma of the Esophagogastric Junction	Phase III multicenter prospective randomized controlled two-arm trial.	**Neoadjuvant CRT** (CROSS)	**Perioperative CT** (FLOT)	438	OS
			RT (41.4 Gy/23 fractions) and concurrent CT with Carboplatin and Paclitaxel (5 weeks) prior to surgery.	5-Fluorouracil, Leucovorin, Oxaliplatin and Docetaxel. Repetition every 2 weeks (d15, q2w). Four neoadjuvant cycles (8 weeks) prior to surgery and four adjuvant cycles (8 weeks) postoperatively are given.		
**Neo-AEGIS** [63]	Esophageal Adenocarcinoma Adenocarcinoma of the Esophagogastric Junction	Phase III multicenter prospective randomized controlled two-arm trial.	**Perioperative CT** (Modified MAGIC or FLOT)	**Neoadjuvant CRT** (CROSS)	366	OS
			Modified MAGIC: 3 cycles of CT pre-surgery and 3 cycles post-surgery. Epirubicin, cisplatin or oxaliplatin and a choice of 5-fluorouracil or capecitabine. Each cycle lasts 21 days. FLOT: 8 cycles of CT in total, 4 cycles of CT pre-surgery and a further 4 cycles of CT post-surgery. Each cycle of CT lasts 14 days/2 weeks.	RT (41.4 Gy/23 fractions) and concurrent CT with Carboplatin and Paclitaxel (5 weeks) prior to surgery.		
**RACE** [65]	Gastroesophageal Junction Adenocarcinoma	Phase III multicenter prospective randomized controlled two-arm trial	**Perioperative CT** (FLOT)	**Perioperative CT + Neoadjuvant CRT**	340	PFS
			Four cycles of neoadjuvant CT with FLOT every two weeks followed by surgical resection 4–6 weeks after day 1 of the last cycle of neoadjuvant therapy.	Two cycles of neoadjuvant induction CT with FLOT. CRT consists of oxaliplatin 45 mg/m^2^ weekly (d1, 8, 15, 22, 29) and continuous infusional 5-FU 225 mg/m^2^ plus concurrent radiotherapy given in 5/week fractions with 1.8 Gy to a dose of 45 Gy over 5 weeks. Resection is performed 4–6 weeks after last treatment with CT/radiation. Adjuvant treatment starts 6–12 weeks after surgery and consists of 4 cycles of FLOT (total treatment period of 26–33 weeks).		
**PROTECT** [26]	Esophageal cancer located under the carena (beyond 25 cm from the incisors) or junctional cancer (Siewert I or II).	Phase II multicenter prospective randomized two-arm trial	**Neoadjuvant CRT** (FOLFOX)	**Neoadjuvant CRT** (Carbo-Paclitaxel)	106	CRR and severe (grade ≥ 3) postoperative morbidity/mortality.
			RT (41.4 Gy/23 fractions) and concurrent every two weeks CT with Folfox scheme (5-Fluorouracil; Oxaliplatin and Folinic acid).	RT (41.4 Gy/23 fractions) and concurrent weekly CT with Carboplatin and Paclitaxel.		

**Table 2 cancers-14-00431-t002:** Design and main characteristics of ongoing trials for post-operative radiochemotherapy in esophageal and gastroesophageal cancer.

Trial Name	Country	Participants	Endpoints	Intervention
Adjuvant radiotherapy, chemotherapy or surgery alone for high-risk histological node negative esophageal squamous cell carcinoma: Protocol for a multicenter prospective randomized controlled trial	China	486 patients: - No prior therapies; - R0 resection; - Thoracic esophgeal squamous cells carcinoma; - pT1b-T4a; - pN0; - High risk features (middle/upper third, LVI/SM, G3); - ECOG PS 0-2; - Adequate organ function.	Primary: DFS. Secondary: OS, adverse events	Experimental groups: ○ Adjuvant CT group: surgery followed by 3–4 week cycles of adjuvant CT with 175 mg/m^2^ paclitaxel and 75 mg/m^2^ cisplatin. ○ Adjuvant radiotherapy group: surgery followed by adjuvant radiotherapy (50 Gy/2 Gy per fr) Control group: surgery alone, without any adjuvant therapy
A phase-II/III randomized controlled trial of adjuvant radiotherapy or concurrent chemoradiotherapy after surgery versus surgery alone in patients with stage-IIB/III esophageal squamous cell carcinoma	China	120 patients: - Age 18–68 years; - Pathologically proven stage-IIB/III esophageal squamous cell carcinoma; - Radical resection (R0); - No prior therapies; - KPS ≥ 70; - Adequate organ function; - No locoregional recurrence or distant metastasis after surgery and before recruitment; - IMRT/VMAT - Adhesion to follow-up	Primary: DFS. Secondary: OS. Other: proportion of patients who complete treatment, toxicity, and out-of-field regional recurrence rate	50.4 Gy/1.8 Gy RT concurrent with paclitaxel (135–150 mg/m^2^) plus cisplatin or nedaplatin (50–75 mg/m^2^) treatment every 28 days. Two cycles will be required for concurrent chemotherapy. 54 Gy/1.8 Gy RT.
Efficacy of Intensity Modulated Radiation Therapy After Surgery in Early Stage of Esophageal Carcinoma; (IMRT)	China	240 patients: - Pathologically proven stage T2-3N0M0 thoracic esophageal squamous cells carcinoma - Radical resection (R0) - KPS ≥ 70; - No prior therapies; - No clear recurrent or metastatic lesions before RT; - IMRT; - Adhesion to follow-up.	Primary: DFS. Secondary: OS.	No intervention: surgery alone. Experimental: surgery plus radiation (50.4 gy/1.8 Gy)
Phase I/II Study of Postoperative Chemoradiation in Patients With Node-positive Esophageal Squamous Cell Carcinoma	China	33 patients: - KPS ≥ 70; - Pathologically proven positive lymph node thoracic esophageal cancer; - Radical resection (R0); - Adequate organ function.	Primary: Maximum tolerated dose of weekly paclitaxel and cisplatin with concurrent RT. Secondary: toxicity, OS, DFS.	Experimental: Arm A Phase 1: weekly paclitaxel (dose escalation) and cisplatin with concurrent RT Phase 2: weekly paclitaxel (dose according to phase 1) and cisplatin with concurrent RT
Phase III Intergroup Trial of Adjuvant Chemoradiation After Resection of Gastric or Gastroesophageal Adenocarcinoma	USA	546 patients: - Pathologically proven stage-IIa/IV M0 stomach/GEJ adenocarcinoma; - En bloc resection; - No prior therapies; - ECOG PS 0–2; - Adequate organ function; - No locoregional recurrence or distant metastasis after surgery and before recruitment; - IMRT/VMAT - adhesion to follow-up	Primary: OS. Secondary: DFS.	Arm 1: leucovorin calcium IV and 5-FU IV on days 1–5 of courses 1, 3, and 4. Courses repeat every 28 days. Concomitant RT and 5-FU IV continuously for 5 to 6 weeks. Arm 2: epirubicin IV and cisplatin IV on day 1 and 5-FU IV continuously on days 1–21 during course 1. Beginning 1 week later, patients undergo RT 5 days a week and 5-FU IV continuously for 5 weeks.
Phase II Study of Postoperative Concurrent Chemoradiotherapy for Esophageal Squamous Cell Carcinoma (ESO-Shanghai 17)	China	74 patients: - Age 18–75 years; - Pathologically proven T3-4N0M0, T1-4N1-3M0 esophageal squamous cell carcinoma; - Radical resection (R0); - No prior therapies; - ECOG PS 0-2; - Adequate organ function.	Primary: LC rate. Secondary: OS.	Experimental Arm: Concurrent CTRT: Paclitaxel 50 mg/m^2^/d, iv over 3 h, d1; Carboplatin AUC = 2 + RT 50.4 Gy/1.8 Gy.

(Abbreviations: LVI: lymphovascular invasion; SM; submucosal metastasis; ECOG PS: Eastern Cooperative Oncology Group performance status; DFS: disease-free survival; OS: overall survival; CT: chemotherapy; IMRT: intensity modulated radiation therapy; VMAT: volumetric modulated arc therapy; KPS: Karnofsky performance status; RT: radiotherapy; GEJ: gastroesophageal junction; CTRT: chemoradiotherapy).

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
