# Peer review of "Modern Management of Esophageal Cancer: Radio-Oncology in Neoadjuvancy, Adjuvancy and Palliation"

_cancers, 2022, doi:10.3390/cancers14020431_

Round 1

Reviewer 1 Report

The authors reviewed the role of radiotherapy in neoadjuvant and adjuvant setting for patients with esophageal cancer. In this area, the ASCO Guideline (ref. 3) seems to be enough and more helpful for readers. No additional scientific information is provided in the current manuscript. While the application of brachytherapy, MR-guided RT, and radiomics was described, more details of definitive chemoradiotherapy rather than palliative approach, combining immune check point inhibitors with chemoradiotherapy in neoadjuvant or definitive setting, and the application of proton beam therapy might be more appropriate and helpful for readers.

In addition, some minor points are needed to be modified.

1) Line 89, nCRT was used without definition. While the authors frequently used RTCT (radiochemotherapy), CRT (chemoradiotherapy) used in line 89 and Table 1 seems to be familiar for readers.

2) In Tables, description of abbreviations is missing.

3) Line 561, The details of reference 20 is missing. → J Clin Oncol. 2014;32(5):385-91

Author Response

Report 1

The authors reviewed the role of radiotherapy in neoadjuvant and adjuvant setting for patients with esophageal cancer. In this area, the ASCO Guideline (ref. 3) seems to be enough and more helpful for readers. No additional scientific information is provided in the current manuscript. While the application of brachytherapy, MR-guided RT, and radiomics was described, more details of definitive chemoradiotherapy rather than palliative approach, combining immune check point inhibitors with chemoradiotherapy in neoadjuvant or definitive setting, and the application of proton beam therapy might be more appropriate and helpful for readers.

Thank you for your comment. We have described more extensively the definitive RTCT approach, reserved for patients unsuitable for surgery, sometimes in a palliative setting. Also, we have discussed more widely the role of proton therapy and immunotherapy.

In addition, some minor points are needed to be modified.

1) Line 89, nCRT was used without definition. While the authors frequently used RTCT (radiochemotherapy), CRT (chemoradiotherapy) used in line 89 and Table 1 seems to be familiar for readers.

Thank you, we have replaced "nCRT" with the previously used acronym "nRTCT".

2) In Tables, description of abbreviations is missing.

We have included the description of abbreviations, as requested by the reviewer.

3) Line 561, The details of reference 20 is missing. → J Clin Oncol. 2014;32(5):385-91

Thank you, we have made the correction.

Reviewer 2 Report

Some phrases may be rewised for a better English language and style .

I suggest you to revise the manuscript because there are some errors, as.

  • line 52 "i.e SCC" but you are describing ADK
  • line 75 ratewith
  • line 89 nCRT i suppose it's nRTCT
  • line 90 bracket open and then not closed

and others...

Author Response

Report 2

Some phrases may be rewised for a better English language and style .

I suggest you to revise the manuscript because there are some errors, as.

  • line 52 "i.e SCC" but you are describing ADK
  • line 75 ratewith
  • line 89 nCRT i suppose it's nRTCT
  • line 90 bracket open and then not closed

and others...

Thank you for pointing this out, we have made the requested corrections

Round 2

Reviewer 1 Report

Generally, the manuscript has been revised properly.

Additional suggestions are regarding the Tables. 

1) While Table 1 and 2 summarized ongoing trials for pre-operative CRT and post-operative CRT, respectively, the format is quite different each other. I suggest that the format of Table 1 might be modified to the format of Table 2.

2) Please explain the meaning of the brackets in Table 1; [3] [6] [8] [9]. The numbers are not match with the reference number. 

3) "Radiation therapy" in Table 1 could be abbreviated to "RT", as in Table 2.

4) In Table 2, "LC" was not described.

Author Response

We do thank the reviwer for the nice comments and for the useful suggestions.

We do reply point-by-point

1) While Table 1 and 2 summarized ongoing trials for pre-operative CRT and post-operative CRT, respectively, the format is quite different each other. I suggest that the format of Table 1 might be modified to the format of Table 2.

--> We do agree and ask the editorial office to format the tables accorging to the journal's style similarly each other

2) Please explain the meaning of the brackets in Table 1; [3] [6] [8] [9]. The numbers are not match with the reference number.

--> we do thank the reviewer, some Ref numbers changed over the manuscript modifications: we fixed the association to the exact reference 

3) "Radiation therapy" in Table 1 could be abbreviated to "RT", as in Table 2.

--> we do thank the reviewer: we adapted the abbreviation as suggested

4) In Table 2, "LC" was not described.

--> yes, we agree. We edited the Table and abbreviation list